# An Overview of Wound Dressing Materials

**DOI:** 10.3390/ph17091110

**Published:** 2024-08-23

**Authors:** Tânia Lagoa, Maria Cristina Queiroga, Luís Martins

**Affiliations:** 1MED—Mediterranean Institute for Agriculture, Environment and Development, University of Évora, Mitra Campus, P.O. Box 94, 7006-554 Évora, Portugal; d50889@alunos.uevora.pt (T.L.); lmlm@uevora.pt (L.M.); 2CHANGE—Global Change and Sustainability Institute, Institute for Advanced Studies and Research, University of Évora, Mitra Campus, P.O. Box 94, 7006-554 Évora, Portugal; 3Department of Veterinary Medicine, School of Science and Technology, University of Évora, Mitra Campus, P.O. Box 94, 7006-554 Évora, Portugal

**Keywords:** skin, wounds, healing, wound dressings, biomaterials

## Abstract

Wounds are an increasing global concern, mainly due to a sedentary lifestyle, frequently associated with the occidental way of life. The current prevalence of obesity in Western societies, leading to an increase in type II diabetes, and an elderly population, is also a key factor associated with the problem of wound healing. Therefore, it stands essential to find wound dressing systems that allow for reestablishing the skin integrity in the shortest possible time and with the lowest cost, avoiding further damage and promoting patients’ well-being. Wounds can be classified into acute or chronic, depending essentially on the duration of the healing process, which is associated withextent and depth of the wound, localization, the level of infection, and the patient’s health status. For each kind of wound and respective healing stage, there is a more suitable dressing. The aim of this review was to focus on the possible wound dressing management, aiming for a more adequate healing approach for each kind of wound.

## 1. Introduction

The skin is the largest organ in the body and the most exposed to aggressions from the external environment, shielding the body from physical, biological, and chemical damage. It protects the body against invasive microorganisms and allergen penetration and contributes to the regulation of body temperature and homeostasis [1,2,3,4]. Wounds are one of the most frequent clinical conditions observed by doctors and veterinarians, often associated with serious clinical complications in human and animal health [5,6]. The development of new drugs and covering materials is under increasing study, aiming to promote a healthy skin state in the shortest possible time. To prevent contamination and dehydration, wounds need effective and simultaneously comfortable covering. It is crucial to avoid the use of antibiotics and antiseptics, both due to growing bacterial resistance and because of the harmful effect of disinfectants on cells, which, by favouring necrosis, promotes bacterial growth. Lately, a large variety of biomaterials have been tested for their antimicrobial and wound-healing properties.

The aim of this review was to systematize the latest developments in wound dressing investigations, focusing on the most popular and promising methods of healing promotion.

## 2. Wound Pathogenesis

The rapid emergence of antimicrobial resistance by pathogens has become an alarming public health problem, leading to chronic infections and high morbidity and mortality rates. For this reason, the focus on the wound treatment process, aiming for a faster healing stage, with less discomfort to the patient, should block contamination, preventing systemic antimicrobial therapy [7].

Recent studies argue that the most suitable materials for wound healing are those that provide a physiological environment at the wound level, preventing contamination and providing physical support. Wound dressings must provide comfort to the patient and promote debridement of necrotic tissue, allowing for the absorption of exudate, facilitating the propagation of oxygen, and providing a moist environment, with minimum toxicity, which stands essential to the healing process [8].

Healing is a dynamic process [2,8], so the choice of the best covering material depends on the healing phase and the type of wound addressed [9]. So far, there is no ideal material that meets all the requirements for all stages of wound healing; therefore, it is necessary to produce a coverage, resulting from the association of several biomaterials [10].

### 2.1. Skin Structure and Its Functions

The skin is the largest organ of animals. It is rather complex, providing an effective defence barrier regarding the external environment, and protecting the body from physical, chemical, and microbial aggressions. The skin also participates in the regulation of body temperature, and the sensitive perception of pain, heat, cold, pressure, and itching [1,2]. Precisely because it is an external and extensive organ, it is exposed to constant aggression, explaining the high number of dermatology cases in human and veterinary medicine, when compared to other systems. This organ also reflects the general state of health, acting like a “mirror of the organism”.

The skin is divided into three layers: epidermis, dermis, and hypodermis [2,3]. The epidermis is composed of several layers, and the deeper ones show a high cell replication rate, being responsible for cell renewal. The newer cells push the older ones towards the surface, undergoing changes in shape and chemical composition, secreting and accumulating keratin, in a process known as keratinization, during which the cells die and form a resistant outer layer. From the most superficial to the deepest, epidermal layers are known as stratum corneum, translucent, granular, spinous, and basal sheet. The dermis is responsible for the greater structural strength of the skin, nutrition of the epidermal cells, and removal of waste products. The hypodermis, also called the subcutaneous layer, is mostly composed of lipid cells, presenting a role as protective padding, an insulator, and an energy reservoir [3].

### 2.2. Wounds and Their Characterization

Wounds may be caused by infection, burns, illness, or accidental damage to the skin, and are frequently associated with pain, disability, and bleeding. Wounds may be classified as open (when there is a break in the skin, leaving the internal tissue exposed) or closed (with no clear disruption of the continuity of the skin, hence without exposure of the underlying tissues). An impaired skin barrier function may lead to an imbalance in skin homeostasis, which may evolve into inflammatory exudative conditions, creating an adequate environment for bacterial, viral, and fungal growth [11,12].

Wounds may be also classified according to their aetiology, duration, degree of contamination, depth, extent, location, and according to specific morphological characteristics [11,13,14]. The healing time will also depend on those factors, allowing for its classification into acute, sub-acute, and chronic. Acute wounds may exhibit various manifestations, ranging from superficial scratches to deeper wounds. It is essential to assure that their evolution does not comprise the formation of a substantial crust extension and that the healing does not extend beyond 3 weeks [14]. Sub-acute wounds will last between 3 weeks and 3 months [13]. Chronic wounds are usually infected wounds, often occurring in immunocompromised patients. These are wounds exposed to factors causing tissue hypoxia and oxidative stress as in diabetes, cardiovascular disease, hypothyroidism, hyperadrenocorticism, and malnutrition with hypoproteinaemia or vitamin and mineral deficiency [15,16,17]. Extensive and deep wounds located in areas of great tension may become chronic as well [14]. Wounds subjected to radiation and hypothermia conditions may also become persistent or chronic [16,18]. Infection represents the highest risk factor for wound healing with Staphylococcus pseudintermedius being the most common bacterium in wound colonization [6]. Infected wounds show a greater chance of developing ischemia, leading to necrosis [6]. A major concern is the development of biofilms, consisting of multicellular bacterial communities, with greater resistance to both antimicrobials and the host’s immune system [13]. Resistance to antimicrobials is frequently related to the dehiscence of sutures with prolonged healing time (3-month minimum duration) and a higher cost of treatment with an increased zoonotic risk [6]; bacteraemia, endocarditis, and bone, joint, and central nervous system (CNS) infections increase [19]. It is, therefore, essential to find an adequate therapeutic approach for the specific classifications of each wound.

### 2.3. Healing Stages

Wound healing (Figure 1) is an inflammatory process mediated by cytokines and growth factors and is a four-stage event: haemostasis, inflammation, proliferation, and remodelling [2]. Briefly, the healing process is marked by an initial inflammatory response in which haemostasis occurs by platelet activation and fibrin formation, followed by the migration of neutrophils, in the acute inflammatory phase, and later by the lymphocytes and macrophages, in the chronic inflammatory phase. A phase of cell proliferation and migration follows, where the migration of keratinocytes from the dermis occurs, and the formation of new blood vessels takes place. Thanks to the displacement of healing mediators, fibroblasts, and keratinocytes, a replacement of the fibrin-rich matrix by granulation tissue arises. Macrophages stimulate fibroblasts to differentiate into myofibroblasts as contacting cells, allowing for the attachment of wound edges. Those cells, along with the fibroblasts, will form the extracellular collagen-rich matrix. In the last phase, there is a reorganization and remodelling of this matrix, and the healing process becomes completed [5,16,17,20,21].

Some traumas require stabilization of the patient, followed by cleaning and management of the wounds, through disinfection with antiseptic solutions, debridement, wound closure, topical antimicrobial treatment, and coverage.

There are different debridement techniques, such as surgical, which involves tools such as scissors, scalpels, and curettes; autolytic, where there is no removal of exudate so that white blood cells and proteolytic enzymes may promote the removal of bacteria and small amounts of necrotic tissue; biological, using larvae and worms to remove dead tissue; enzymatic, through the use of proteolytic enzymes such as Streptokynases, trypsin, proteinases, and collagenases, to remove devitalized tissue and destroy bacterial biofilm; and mechanical, where force is involved to remove tissue or gauze adherent to the wound. According to Kietzmann, debridement is increasingly recommended to remove necrotic tissue, without resorting to antiseptics, as these may promote necrosis, fostering the growth of bacteria, and delaying the normal healing process [22]. Silvestro et al. [17] also argued that topical antimicrobials could delay the healing process and should only be used when there is confirmation of infection or in cases of immunosuppressed animals [17].

## 3. Types of Covers

Wounded skin should be coated to minimize damage and the risk of infection, decrease available oxygen, reduce pH, and promote restoration integrity of the damaged tissue as quickly as possible. In the last decades, many biomaterials have been developed and introduced in the market, aiming to act as skin substitutes [23].

Among the available dressings, the choice of the ideal dressing will depend on the type of wound, its depth, location, and extent, as well as the degree of exudation, the presence of infection, and the level of adhesion to the wound, of the dressing material. Traditional cotton and gauze bandages, for instance, are able to absorb large amounts of exudate, which, in turn, may cause wound dryness and pain when detaching the dressing, leading to a slower healing process. Generally, a coating with a high water vapour transmission rate is more likely to lead to wound dehydration, while a coating with a poor transmission rate tends to allow for the accumulation of exudate. The presence of exudates in the wound promotes maceration and excoriation of the surrounding skin, and poor cohesion of the newly formed tissue layers, slowing down the healing process [9,24,25]. The dressings should be soft and flexible to allow for easier and painless applications and removals. Dryer environments favour cell adhesion [9].

An ideal coating (Figure 2) must therefore have different properties, including being inexpensive, non-antigenic, durable, flexible, non-toxic, non-adherent, easy to place on lesions, applicable in a single operation, easy to remove, non-traumatic, composed of biological material (which requires minimal processing), not allowing water loss, adaptable to irregular lesion surfaces, allows for gas exchange and the removal of excess exudate, offers mechanical protection, protects from contamination, being biodegradable, and enhances the skin-inherent healing process. Coatings comprising biological material also require less processing with fewer adverse reactions [2,8,9,10,14]

Dressings are classified according to their antimicrobial role, their absorbent and occlusive action on the wound, or according to their ability to adhere and promote debridement. Covers allowing for debridement contain hypertonic solutions such as sugar or honey, polyhexanides, silver salts, copper tripeptide complex, acemannan, resulting from the fermentation of aloe vera, platelet derivatives, chitosan, or maltodextrin (oligomers from glucose).

According to the nature of the material they are made of and their physical forms (Figure 3), dressings can also be classified into three categories: biological (from animal or plant origin), biological–synthetic, or just synthetic. According to its physical form, available coatings can be oils, films, foams, hydrogels, hydrocolloids, or membranes [26]. Ndlovu et al. [24] grouped dressings into four groups (Table 1): traditional, synthetic, biological dressings or skin substitutes, and bioactive [22].

### 3.1. Traditional Dressings

Traditional dressings (e.g., gauzes and cotton composites, among others) are primarily used at an early stage to promote haemostasis and to clean and dry wounds [28]. This type of dressing is affordable but presents several limitations such as poor permeability to gases and the damaging of newly formed epithelium. Their fibres stick to the granulation tissue, causing pain when removing the dressing, eliminating the new tissue formed and delaying the healing [24,29].

### 3.2. Synthetic Dressings

Synthetic dressings are usually based on synthetic polymers and biopolymers. Synthetic coatings include hydrogels, hydrocolloids, paraffin gauzes, foams, semi-permeable films, and silicone blends [15].

#### 3.2.1. Foams

Foams are made of solid materials with a porous matrix, usually composed of polyurethane and, sometimes, a layer of soft silicone. Polyurethane-based products are covers mostly used for their biocompatibility, favourable mechanical properties, low toxicity, softness, flexibility, gas permeability, and moisture retention [14]. The polyurethane sponges have a good absorption capacity, being useful in wounds with higher levels of exudation. These non-adherent materials are particularly adequate for promoting the formation of healthy granulation tissue.

#### 3.2.2. Films

Semi-permeable films are thin, flexible, semi-occlusive, transparent, porous, and non-absorbable coatings. They allow for the transmissibility of water vapour and the oxygenation of the wound and provide a barrier against external contamination. These coatings are indicated for wounds with little or no exudate. Their use is indicated in the final stage of healing to promote epithelialization. In hairy body sites, its adhesion is compromised, and the removal of the cover is usually traumatic [14].

#### 3.2.3. Hydrogels

Among the various coatings available, hydrogels are the class of materials that best mimic the structure of the extracellular matrix of tissues, due to their ability to retain large amounts of water and their soft consistency [53]. Hydrogels are usually transparent and consist of polymeric networks, capable of absorbing large amounts of water or biological fluids, with simultaneous moisturizing, allowing for tissue rehydration [8,54]. These materials have different applications in medicine or the pharmaceutical industry. Hydrogels allow for good biocompatibility and anti-inflammatory activity, promote cellular immunological activity, owing to low toxicity and low allergenicity, and are easy to remove from the wound [1,8,54]. They present excellent tissue adhesion, allowing for not only a stable protective barrier but also interaction with different wound components and surrounding tissues. Moreover, hydrogels favour the cell adhesion, vascularization, migration, and proliferation of fibroblasts and other growth factors. Hydrogels shorten the healing period by reducing the formation of granulation and necrotic tissue through their autolytic activity [54] and also assist in the delivery of the nutrients and oxygen required for the healing process [1,8,55]. However, the highly hydrated environment may facilitate microbial colonization. Hydrogels are also used for controlled drug delivery systems, which may be additionally useful for wound management [56].

Hydrogels can be considered reversible or physical gels, based on their polymer arrangement, crosslinking procedure, and external stimuli like temperature and pH. These networks are held together by molecular bonds, and/or secondary forces such as ionic, hydrogen bonding, and hydrophilic forces [8,57]. On the other hand, hydrogels can be permanent or chemical when the networks are covalently linked. The polymers used to synthesize hydrogel networks can be divided into two groups, according to their synthetic or natural origin. The synthetic polymers most commonly used to prepare hydrogels are polyethylene glycol and polyhydroxyethyl methacrylate. Besides being easily reproducible, the use of synthetic polymers generally results in hydrogels with high mechanical strength [57]. Comparatively, hydrogels from natural polymers easily lose their characteristic viscosity due to the destruction of the molecular bonds responsible for keeping the network/matrix intact, showing critical biodegradability [1,49]. Some examples of natural polymers giving rise to hydrogels are dextran, agarose, hyaluronic acid, collagen, alginate, gelatine, cellulose, starch, chitosan, and xanthan [55]. Some associations such as chitosan with gelatine are especially useful as the resulting hydrogel presents increased resistance to degradation by collagenases [1,8].

#### 3.2.4. Hydrocolloids

Hydrocolloids originate from a mixture of colloidal compounds with elastomers and alginates. These mixes are generally biodegradable and biocompatible and stimulate autolytic debridement, also favouring inflammation, angiogenesis, collagen synthesis, and epithelialization [18,58]. Hydrocolloids are not suitable for deep wounds, especially if infected, as they reduce the amount of available oxygen required for healing. These formulas are able to absorb minimal to moderate amounts of exudate and inhibit bacterial multiplication by lowering the pH [18,58]. They contribute to pain control [30,31] but may inhibit wound contraction in the area where the skin adheres to the dressing, limiting its use in the final phase of healing [32,33].

### 3.3. Biological Dressings and Skin Substitutes

Different research studies are currently in the process of being dedicated to the production of new coatings through the synthesis and modification of biocompatible materials [23]. These coatings may originate from biopolymers such as collagen, gelatine, alginate, chitosan, dextran, cellulose, and elastin [53] and present relevant advantages such as permeability to water vapour and oxygen, high biocompatibility, non-expensive, highly durable, with good ability to induce tissue regeneration, the prevention of microorganism colonization, and trauma [59].

Several skin substitutes (SSs) are commercially available, being used for different clinical purposes [60]. The SSs, mainly composed of collagen, may function as autografts, but despite being inexpensive, limitations have been shown, associated with the risk of infection and disease transmission [24,32,60]. Therefore, the good use of biological dressings and skin substitutes should be limited to low-exuding wounds [24,53].

#### 3.3.1. Collagen

The main coverings of animal origin are essentially collagen-based. Collagen is the most abundant protein in connective tissue and can be extracted from bovine or porcine skin, tendons, intestine, or bladder mucosa, or from rat tail [32]. This collagen from natural sources is quite versatile, allowing for processing (extraction, purification, and polymerization) to undergo physical changes (under high pressure and low temperature), chemical modifications (by the addition of detergents and high osmolarity solutions of, or chelation with, EDTA, and basic or acid treatment), and enzymatic transformations (by trypsin digestion) giving rise to several biomaterials, with different applicability. It is also possible to associate other biomaterials with collagen, like elastane, chitosan and glycosaminoglycans. From collagen processing, it is even possible to obtain sponges, films, membranes, spheres, and hydrogels [18,32]. Collagen participates in the process of haemostasis and tissue repair, and it is biocompatible, biodegradable, has low allergenic power, is non-toxic, and shows high tensile strength [32]. Commercial collagen sponges are useful for burn and ulcer dressings as they have high porosity and high water absorption capacity. Several studies evidenced that collagen-based dressings promote cellular recruitment, activate the inflammation phase of wound healing, and stimulate the formation of new granulation tissue and epithelial layers. Macroscopically, studies carried out in vitro and in vivo with rabbits revealed great stability of these materials to thermal variations and in the presence of oedema. However, microscopically, focal necrosis was detected, both in the epidermis and dermis [18]. Collagen may also be extracted from marine invertebrates and plants such as undersized fish, jellyfish, sharks, starfish, and sponges. These organisms are promising sources of collagen because there are no religious limitations for its use, and there are no reports of possible transmissible diseases [61]. However, collagen of fish origin shows limitations related to less crosslink with its mammal equivalent and reduced mechanical strength, resulting in difficulty in preserving this type of material [18,21,61]. Collagen-derived products find wide applications as membranes, bone graft material, agents for local drug delivery, and haemostatic agents. Liposomes can be added to collagen gels, under encapsulation and may be used for the controlled release of insulin, and growth factors, among other drugs. There are also associations of collagen with alginate to promote the inflammatory and healing phases of wound healing [32].

#### 3.3.2. Gelatine

Gelatine results from the denaturation of collagen obtained from the skin and bones of animals. This material offers good biocompatibility, shows haemostatic action, low toxicity, and antigenic properties, is biodegradable, and presents good plasticity and adhesion capacity [32]. Gelatine shows poor mechanical and antimicrobial properties [24]; however, associating sodium alginate with gelatine will allow for greater elasticity with less resorption power and toxicity [33,34].

#### 3.3.3. Cellulose

Cellulose is a biopolymer, mostly obtained from wood, and is also used in the paper industry, due to its high availability. Cellulose may come from plants, or be produced by algae, fungi, and some species of bacteria. Cellulose brings numerous advantages such as high porosity, permeability to liquids and gases, and high water holding capacity, and it is biodegradable, non-toxic, and soft. Bacterial cellulose has good biocompatibility when compared to plant cellulose and provides extracellular protection for its production of bacteria, through a kind of biofilm that works as a protective barrier [25]. It is suitable for wound dressing due to its capacity to enable autolytic debridement, relieve pain, and promote the formation of granulation tissue. Portela et al. [25] compared different treatment approaches for the healing of wounds with similar characteristics and about 15 mm in diameter. The authors reported the highest healing rate in wounds treated with mesenchymal stem cells, in association with bacterial cellulose, followed by the ones treated with a bacterial cellulose hydrogel, both with better results than the control group [25].

#### 3.3.4. Bamboo

Plants have been crucial as natural medicines, both for local application and the development of therapeutic foods. Bamboo fibres contain cellulose, lignin, hemicellulose, and extractives identical to other natural fibres [62,63] Bamboo medical textiles are used for the controlled release of herbal extracts and show potential antimicrobial, anti-inflammatory and wound-healing properties. These fibres enhance wound contraction and increase epithelialization [64]. Herb-based dressings are non-toxic, allowing for use for long periods. Absorptive capacity is determined by its hydrophilic abilities and porosity. Dressings with bamboo provide high strength, besides economic viability and sustainability extraction, due to its environmental-friendly nature and rapidly growing properties [62,63,65].

#### 3.3.5. Hyaluronic Acid

Hyaluronic acid is a polysaccharide, belonging to the glycosaminoglycan family, and is a key component of joint synovial fluid, cartilage, the vitreous humour of the eye, and the skin of vertebrates. It is involved in the inflammatory response, angiogenesis, cell adhesion, and the regeneration process. Thanks to the intrinsic properties of hyaluronic acid, such as biocompatibility, biodegradability, and hydrophilic character, it is used to produce various types of coverage like hydrogels, sponges, and films [49,66].

#### 3.3.6. Sodium Alginate

Sodium alginate is a natural fibrous polymer derived from brown algae. When compared to synthetic coatings it has some shortcomings such as poor thermal stability, high hydrophilic power, and poor mechanical properties [20,67]. Alginate forms a gel when it comes in contact with wound exudate, and can absorb twenty times its weight, due to the high porosity. As alginate needs moisture to improve its performance, it is not suitable for dry wounds. Hence, to prevent excessive wound desiccation, alginate dressings should have a second layer of foam or hydrocolloid, which should not be occlusive in case of infection [18]. Thanks to the presence of calcium in its composition, it is useful for clot formation and consequently for the haemostatic process. Ambrogi et al., in 2020 [68], developed alginate films with silver nanoparticles, presenting good moisturizing properties, and, due to the slow release of silver nanoparticles, it was shown to be able to promote antimicrobial and anti-biofilm activity [68].

#### 3.3.7. Extracellular Matrix Bands

Extracellular matrix bands are acellular, biodegradable, and sterilized sheets, most commonly made from porcine small intestinal submucosa or urinary bladder submucosa matrix. These bands provide structural proteins, growth factors, cytokines, and their inhibitors in physiological proportions, allowing for the stimulation of angiogenesis and presenting antibacterial properties. However, this product requires specific techniques for application, such as well-prepared bedding with debridement, no other topical medication or cleaning agents, and no exudates. Drainage is essential, demanding fenestrated band material when exudation is relevant [35,36].

#### 3.3.8. Omentum Flaps

The greater omentum or epiploon is a large, flat layer of adipose tissue covered by the visceral peritoneum. The greater omentum originates from the greater curvature of the stomach and hangs down like an apron, floating over the surface of the intra-peritoneal organs, including the small and large intestines. It is a versatile and essential component of the abdominal cavity, playing vital roles in protection, fat storage, and immune response. The lesser omentum extends from the liver to the lesser curvature of the stomach and the beginning of the duodenum and contains ligaments, vessels, and lymph nodes [38]. Omentum flaps are used to cover tissue defects, promote a vascular bed, control adhesions, and fight infections. These materials allow for the establishment of circulation and exudate drainage [5]. The omentum also aids in wound healing and regeneration due to its unique milky spots that produce immunomodulatory cells. Activated omentum stromal cells secrete cytokines that promote tissue repair. Recent research by Liu et al. [37] showed that omental-derived cells may help reduce open epidermis length and increase epidermal cell layers in burn wounds, underscoring the omentum’s regenerative capabilities [37]. Laparoscopically harvested omentum flaps were effectively used to reconstruct extensive and complex wounds, providing well-vascularized, pliable tissue with reliable vascular anatomy [37]. 

#### 3.3.9. Autologous Platelet-Rich Plasma

The autologous platelet-rich plasma favours epithelialization, contraction, and neovascularization in the healing process, and, being autologous, it does not present biological risks concerning the transmission of diseases. Autologous platelet-rich plasma is usually used in wounds with associated vascular problems and therefore with a high rejection rate. Gupta et al. [51] followed two groups of patients who presented chronic wounds and were subjected to skin grafts. In one of the groups, the autologous platelet-rich plasma was injected daily, showing significant improvements in the graft uptake rate, when compared to the control group, undergoing a standard method [69].

### 3.4. Bioactive Wound Dressings

Bioactive dressings can be used for the transport and delivery of drugs (antimicrobial or anti-inflammatory agents, growth factors, and other bioactive agents) to specific tissues or organs at the right time [24,26,70]. In drug delivery systems, encapsulated therapeutics can be released through various mechanisms, widely classified as active or passive delivery. In active delivery, the release is triggered by environmental stimuli (such as pH, temperature, enzymes, chemical reactions, redox reactions, etc.) or external stimuli (such as magnetic fields, electric fields, light ultrasound, etc.). In contrast, passive delivery relies on the diffusion of the drug through the carrier matrix into the surrounding medium from the high-drug-concentration region to the low-concentration region (such as blood). The rate and extent of drug absorption depend on several factors, such as the route of administration, the physicochemical properties of the drug, the type of formulation, and drug–food interactions [71,72]. These systems reduce systemic toxicity and deliver maximum drug levels directly to the target site. The main materials used for controlled and targeted drug delivery are poly (lactic-co-glycolic), poly (vinylpyrrolidone), poly (vinyl-alcohol), poly (hydroxyalkylmethacrylates), hydrocolloids, alginate, hyaluronic acid, and collagen [24]. Different agents such as antimicrobials, nanoparticles, and natural products (essential oils, honey, propolis, and chitosan) have been incorporated within these structures allowing for improved antimicrobial properties [26].

#### 3.4.1. Curcumin

Curcumin is the main bioactive component present in the rhizomes of Curcuma longa Linn (turmeric) [39]. Curcumin is a bioactive phenol exhibiting anti-inflammatory, antioxidant, tissue-protective, chemoprotective, antiviral, and immunomodulatory properties. Curcumin also presents a broad spectrum of antibacterial actions including bacterial membrane disruption and disruption of biofilm formation and exerts potent antioxidant effects by scavenging reactive oxygen species (ROS) [39]. Studies in animals revealed low toxicity, demonstrating good tolerability of daily doses between 4000 and 8000 mg [44]. Curcumin promotes cell proliferation and migration by activating growth factor signalling pathways, such as the epidermal growth factor receptor (EGFR) pathway, and modulating the expression of the extracellular matrix (ECM) components [44]. Fu et al. [40] reported that mice treated with curcumin-loaded poly (ε-caprolactone)-poly (ethylene glycol)-poly (ε-caprolactone) (PCEC) fibres exhibited faster re-epithelialization and blood vessel formation as well as an increase in collagen deposition over time [40,44]. Ranjbar-Mohammadi et al. [45] demonstrated that curcumin-loaded poly (ε-caprolactone)/gum tragacanth (PCL/GT/Cur) electro-spun fibres increased collagen content and accelerated the wound healing process in diabetic male Sprague-Dawley rats [44,45]. In a comparative study utilizing a skin incision model, it was observed that curcumin facilitated complete epithelial repair, promoted angiogenesis during the proliferative phase of wound healing, and accelerated wound closure [43]. Curcumin possesses potent anti-inflammatory properties by inhibiting the activity of pro-inflammatory mediators, such as cyclooxygenase-2 (COX-2), interleukin-1 (IL-1), interleukin-8 (IL-8) tumour necrosis factor-1 (TNF-1), matrix metalloprotease 9 (MMP-9), and matrix metalloprotease 3 (MMP-3) [43]. The ability of curcumin to inhibit the function of NF-(κ)B (nuclear factor kappa B), a regulatory protein central to inflammatory response initiation, is also remarkable [43]. By attenuating inflammation, curcumin promotes a conducive microenvironment for optimal wound healing and falls under class IV of the biopharmaceutical classification system, characterized by low solubility and poor permeability [39,43,44]. These attributes contribute to diminished absorption, limited bioavailability, and stability concerns. Additionally, curcumin undergoes rapid metabolism at specific intervals. To address these challenges, various formulations have been explored, such as nanoparticles, liposomes, nanogels, nano-emulsions, and the use of adjuvants and nanocrystals, reporting promising results in promoting wound healing in patients with acute wounds, chronic wounds, and burns [43,73].

#### 3.4.2. Chitosan

Chitosan is a naturally occurring glycosaminoglycan, derived from the exoskeleton of crustaceans, molluscs, insects, and the cell walls of some fungi. Obtained through the alkaline deacetylation of chitin, it is biodegradable, biocompatible, and has a good function as an antimicrobial agent (bactericidal, bacteriostatic, fungicidal, and fungistatic) [74,75,76]. Studies state that chitosan favours haemostasis and promotes the infiltration and migration of polymorphonuclear neutrophils and macrophages [75]. In trials with diabetic rats, chitosan accelerated the healing phase, allowing for increased migration of fibroblasts and the deposition of collagen and growth factors in their matrix, accelerating the speed of the healing and maintenance of homeostasis [13,77]. This material has osteoinductive and cartilage-inducing activity as well as non-toxic, mucoadhesive, non-carcinogenic, immunostimulant, biocompatible, bioresorbable, and bioactive properties [74,75]. However, it is susceptible to structural changes due to two reactive groups—hydroxyl and amino [78]. The free binding of the amino group is also related to its antibacterial properties [10,79]. Besides the microbial agent present and the cell age, the effectiveness of chitosan action also depends on the solvent used, the concentration of chitosan and its degree of deacetylation, the presence of a plasticizer, hydrophilic/hydrophobic characteristics, influencing its solubility, chelating effects, and the mixing process used. Although owning a broad spectrum of antimicrobial activity, chitosan exhibits differing inhibitory efficiency against different fungi, viruses, and Gram-positive and Gram-negative bacteria. The mode of action of chitosan against microorganisms can be classified based on extracellular effects, intracellular effects, or both [47,51,79,80,81]. The chelate effect of amino groups can cause electrostatic attraction of anionic compounds, such as lipids, cholesterol, metal ions, proteins, and macromolecule anionic residues, by altering their normal function [51,77,80,82]. This activity is affected by pH, being higher at a lower pH (4.5–5.9). Diluted with acidic solutions, the positive charges of chitosan compete with Ca^+^ for electronegative sites on the membrane, conferring instability, weakness, and increased permeability of the cell membrane, leading to cell death [48,82].

#### 3.4.3. Xanthan Gum

Xanthan gum is a high molecular weight natural polysaccharide, obtained from the bacterium Xanthomonas campestris after a fermentation process [55,83]. It has good rheological properties and works as a stabilizer for water-based products, allowing for their retention. Ilomuanya et al. [83] evaluated the effectiveness of various formulations of a xanthan-rich hydrogel, and different concentrations of hyaluronic acid and silver sulfadiazine, concluding that the incorporation of 0.1% silver sulfadiazine and 1.5% hyaluronic acid in the xanthan-rich hydrogel led to the best results in terms of healing rate after 14 days.

#### 3.4.4. Nanomaterials

Gold, silver, copper, zinc, titanium, and magnesium nanomaterials are used for their bioavailability, absorption capacity, mucoadhesive, and protective properties, and as antimicrobial therapy [8]. Nanomaterials allow for the controlled release of drugs trapped in capsules or through the surface membrane [7]. Silver nanoparticles are the most used as they present less toxicity to patients’ cells and greater antibacterial effect against Gram-positive and Gram-negative [84], antiviral, and fungicidal activity [85]. Silver affects microbial plasma membrane proteins, interfering with transmembrane transporting systems and causing “perforation” in their structure, with the dissipation of H^+^ ions and consequently cell death [7]. Diniz et al. [67] followed the evolution of wounds in Wistar rats for 14 days and concluded that wounds covered with hydrogel rich in silver nanoparticles produced new granulation tissue and developed collagen-rich crusts in a shorter time than the control group [67]. Silver nanoparticles can be made available in the form of hydrocolloids, hydrogels, and alginates. The disadvantage of using these nanoparticles is the difficulty in controlling their shape, size, and stability as they oxidize easily and tend to form aggregates [48]. Furthermore, in higher concentrations, heavy metals are toxic to tissue cells [86,87], having shown in vitro cytotoxicity against keratinocytes and fibroblasts and, consequently, delaying healing in vivo [87]. For this reason, slow-release systems for nanoparticles are crucial. The degree and nature of polymer matrix bonds are key issues for the regulation of nanoparticle release.

Electrospinning is the most effective technic for the fabrication of nanofibres, manipulating the product’s shapes and developing novel drug delivery systems. This technique increases nanometre porosity, transforming one-dimensional nanofibres into a three-dimensional sponge structure, allowing for gas circulation and efficient drainage of exudate without increasing the risk of contamination, and qualifying the purposeful control of the loading substances. A wide variety of natural and synthetic polymers are used for electrospinning [88,89,90].

#### 3.4.5. Essential Oils

Essential oils (EOs) are obtained from plant extracts and have antimicrobial activity against Gram-positive and Gram-negative bacteria (*Origanum vulgare* and *Melaleuca alternifolia*) [91], fungi (*Mentha piperita*, *Lavandula pedunculata*, thyme, and rosemary) [92], and viruses (*Eucalyptus globulus* and *Lavandula officinalis*) [93], in addition to antiparasitic (peppermint, lemon, and tea tree EO) [94], insecticidal, anti-inflammatory, and immunostimulant properties. These oils present antioxidant action, stimulate blood circulation at the skin level, reduce fluid retention, and can reduce the development of bedsores and wrinkles [19,95]. However, as they are highly lipophilic components, the poor solubility in aqueous media renders contact and permeabilization of cell membranes difficult [96]. Other studies proved the beneficial action of combinations of EOs (in the form of spot-on or shampoos) as a new strategy against bacterial infections, allowing for the restoration of the cutaneous barrier and the characteristic microbiome of the skin, and also preventing the use of antibiotics and the resulting development of antimicrobial resistance [91,97,98,99]. Furthermore, essential oils have an anti-inflammatory activity, by inhibiting the production of tumour necrosis factor and interleukins, promoting angiogenesis and consequently accelerating healing [15]. According to Elaine dos Santos [50], it might be worth using hydrogels with the association of EOs and nanoparticles, or nano-emulsions to increase the stability of the compound [50]. However, these materials have inherent disadvantages because they fight both microbial and the patient’s cells [7]. 

As with vegetable oils, Omega-6 fatty acids also allow for the maintenance of a moisturized environment in the wound, accelerating angiogenesis, and preventing dehydration and local tissue death. Furthermore, essential oils also promote autolytic debridement and reduce local pain [91].

#### 3.4.6. Honey

Honey is a supersaturated solution of natural sugar, with about 17% water content, fructose, glucose, maltose, sucrose, and other types of carbohydrates. The biological variety of honey depends on the floral resources. Not all honeys are equally effective in healing wounds. The physical characteristics of Manuka Honey, such as high osmolarity, acidity, and an enzyme composition that produces hydrogen peroxide and methylglyoxal (MGO) (non-hydrogen peroxide components), contribute to its antimicrobial [84,100], anti-inflammatory, immunostimulant, and antioxidant action, alone or associated with conventional therapy [8,10,101,102]. Bulman et al. [100] incorporated MGO in polyvinyl alcohol fibre compositions and confirmed bactericidal activity against *E. coli* and *S. aureus* [26,100]. Some authors believe that the acidic pH of honey may enhance macrophages’ killing of bacteria and inhibit microbial biofilm formation. Additionally, the high osmolarity of honey provides an unfavourable environment for microorganism survival and growth. Furthermore, hydrogen peroxide forms a highly toxic hydroxyl radical, which is involved in microbial killing. Many studies prove that the exposure of E. coli to low concentrations of hydrogen peroxide results in DNA damage that causes mutagenesis and kills the bacteria [26,103].

#### 3.4.7. Propolis

Propolis is a natural resinous product produced by honeybees, using substances collected from plants, which are changed by the bee’s salivary secretion. Propolis contains flavonoids, terpenes, phenolic acids, aldehydes, and ketones, and promotes tissue healing, by granulation growth and limitation of scar formation. Many studies have confirmed that propolis contains bioactive compounds with antibacterial [104], antiviral [105], antifungal [105], and anti-biofilm activity [104]. The exact way propolis fights off microorganisms is still not entirely clear [106]. It seems to work through a combination of its various components, working together, rather than one specific mechanism. However, certain species of microorganisms are more vulnerable to propolis than others. Some research suggests that propolis might damage the structure of microorganisms, hampering their normal metabolism This natural substance seems to have multiple targets regarding bacterial cells. By creating a physical barrier and interfering with enzymes and proteins that are necessary for the bacterial invasion process, propolis seems to hinder microorganisms, blocking their ability to replicate and preventing them from invading host cells. Additionally, propolis disrupts the metabolic processes of microorganisms by interfering with cellular structures and components responsible for energy production [106]. Numerous in vitro and in vivo studies have investigated the effects of propolis on different aspects of the wound healing process. In the inflammatory phase, propolis has been shown to modulate immune responses, reduce inflammatory cytokine levels, and promote M2 macrophage polarization, leading to accelerated wound closure and resolution of inflammation. Propolis has analgesic and antioxidant action after acute trauma and burn [104,105,106,107,108]. It inhibits the occurrence of reactive oxygen species and consequent tissue damage [109]. In the proliferation phase, propolis stimulates fibroblast proliferation and collagen synthesis, enhances angiogenesis, and accelerates re-epithelialization, thereby promoting tissue regeneration and granulation tissue formation. In the remodelling phase, propolis facilitates the remodelling of newly formed tissue, promotes wound contraction, and improves the tensile strength of healed wounds. Ibrahim et al. [110] tested a sponge matrix from collagen hydrolysates and honey–propolis wax (HPW) on wound healing, in a mouse model, revealing that HPW increased the re-epithelization and the overall wound healing rate in mice [110].

## 4. Conclusions and Future Perspectives

The skin is the biggest organ of the body, and, consequently, wounds are the most frequently attended conditions in veterinary medicine. In a growing urban lifestyle, with a higher prevalence of obesity and diabetes, wound handling becomes vital both for humans and companion animals. The longer the wound takes to heal, the higher the cost, the greater the morbidity, and the higher the zoonotic risk. Furthermore, the use of antibiotics increases the selection for resistant bacteria. Topical adjuvant therapies with dressings made from natural products, new materials, innovative production techniques, and newer strategies for active principal delivery within the internal environment, focusing on the healing performance, stand as a permanent motivation. Aiming at the improvement of health and quality of life, by decreasing infection, pain, and time of healing, with a consumption of drugs as reduced as possible, also having in mind the cost of treatment and the prevention of biofilm formation, wound healing should stand as the focus with the current main concern of avoiding the increasing antimicrobial resistance. 

Despite the current advances in wound treatment strategies, several limitations still arise and are identified as key points for the next studies. In fact, as wound healing processes are very complex considering different cellular and molecular mechanisms, a single treatment is often not sufficient to experience a satisfactory wound healing process, the rational decision is that different methods should be combined when possible.

The future of wound dressing materials is also rooted in the science of cutting-edge materials, biotechnology, and digital health. Advances in smart bioactive and biodegradable materials, along with tailored and precise medical approaches, are poised to transform wound care. The fusion of these technologies with digital health platforms will improve monitoring, patient involvement, and overall treatment results, making wound care more effective, personalized, and widely accessible. 

We believe that this review may result in being truly useful for choosing the right wound-handling procedures to be implemented on each wound, depending on its characteristics.

## Figures and Tables

**Figure 1 pharmaceuticals-17-01110-f001:**
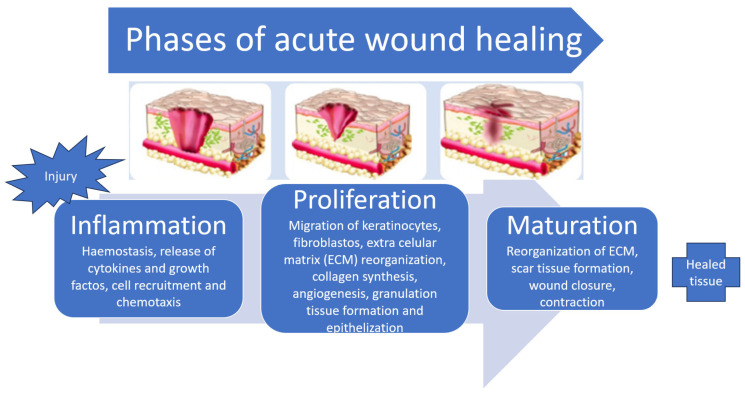
Distinct phases of wound healing (adapted from [14,18]).

**Figure 2 pharmaceuticals-17-01110-f002:**
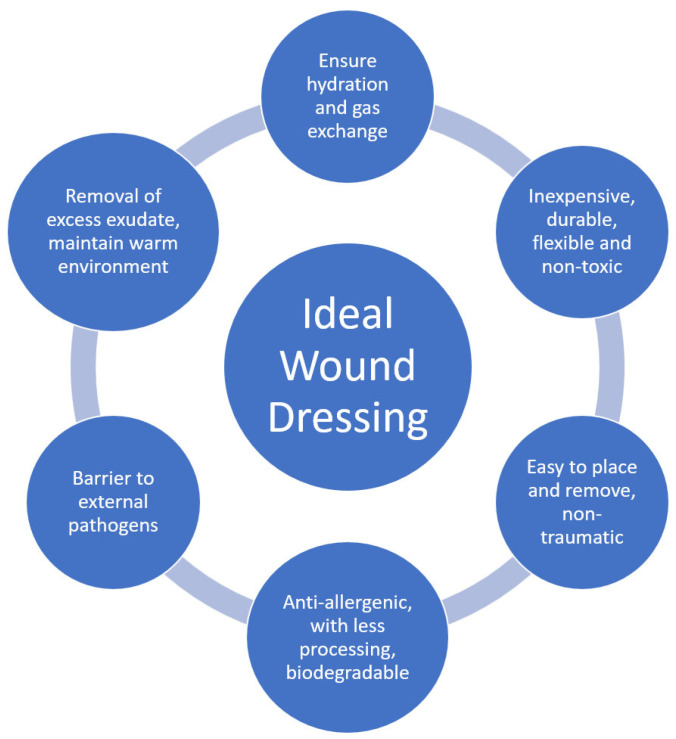
The ideal wound dressing features (adapted from [18]).

**Figure 3 pharmaceuticals-17-01110-f003:**
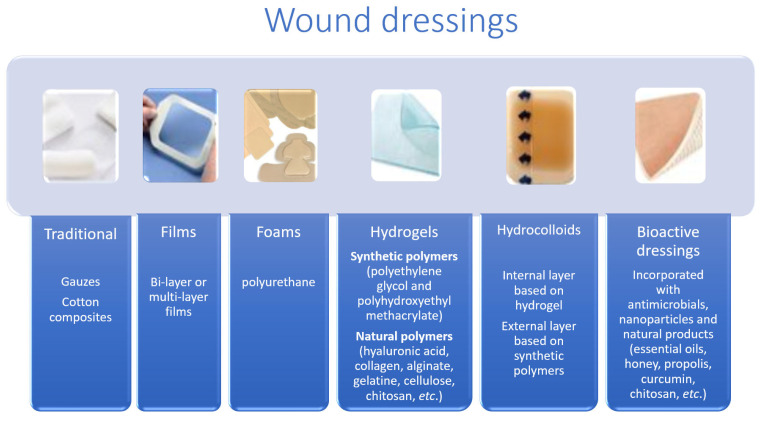
Wound dressings: traditional, films, foams, hydrogels, hydrocolloids, and bioactive dressings (adapted from [27]).

**Table 1 pharmaceuticals-17-01110-t001:** Characteristics and applications of the dressings.

Wound Dressing	Characteristics	Applications and Details	References
Traditional Bandages, gauze, and cotton composites	Poor permeability to gases, prevent bacterial contamination; absorb exudates and fluids; low cost, adherent, painful removal.	Used in early stage of healing, in clean and dry wounds or with mild exudate levels.	[24,28,29]
Synthetic Dressings			
Foams	Soft, non-adherent, and flexible, with gas permeability and moisture retention. Absorb large amounts of fluids and ensure thermal insulation.	Suitable for cavity wounds and ulcers.	[14,30]
Films	Flexible, semi-permeable, semi-occlusive, porous, and non-absorbable.	Indicated in final stage of healing, in dry or low exudative wounds.	[14,30]
Hydrogels Synthetic or natural origin (dextran, agarose, hyaluronic acid, collagen, alginate, gelatine, cellulose, chitosan, and xanthan)	Facilitate autolytic debridement, anti-inflammatory activity, promote cellular and immunological activity; easy to remove.	Suitable for dry, low, or moderate exudative wounds; chronic wounds: necrotic, ulcers, and burn lesions.	[18]
Hydrocolloids	Autolytic activity, favouring inflammation, angiogenesis, collagen synthesis, and epithelization; great adhesion property and occlusive; inhibit bacteria growth; reduce oxygen availability.	For wounds with low to moderate drainage, minor burns, and traumatic wounds; not indicated for ulcers or highly exudating wounds; limited use in final phase because they inhibit wound contraction.	[18]
Biological Dressings and Skin Substitutes			
Animal origin: Collagen	High porosity, high water absorption capacity; promotes cellular recruitment, activates inflammation phase and formation of new granulation tissue.	For burn and pressure ulcers; exudative wounds.	[31,32]
Gelatine	Haemostatic action; good plasticity and adhesion capacity; poor mechanical and antimicrobial properties.	For burn and pressure ulcers; exudative wounds.	[33,34]
Herbal origin: Cellulose	High porosity, permeability to liquids and gases, autolytic debridement, relieves pain, and promotes granulation tissue.	For burn and pressure ulcers; exudative wounds.	[25]
Extracellular matrix bands	Stimulate angiogenesis and have antimicrobial properties; require well-prepared bedding with good debridement.	Dry wounds (without exudates) or fenestrated band when exudation is relevant; drainage is important.	[35,36]
Omentum flaps	Favours epithelization, contraction, and neovascularization.	Wounds with vascular problems and high rejection rate.	[37,38]
Bioactive Dressings			
Curcumin, chitosan, essential oils, xanthan gum, honey, and propolis	Anti-inflammatory action, antioxidant, tissue-protective, chemoprotective, antiviral, immunomodulatory properties, antimicrobial and anti-biofilm.	Indicated for acute and chronic wounds like ulcers and infected lesions.	[10,18,39,40,41,42,43,44,45,46,47,48,49,50,51]
Nanomaterials (silver, copper, gold, etc.)	Promotion of angiogenesis, antimicrobial properties, biofilm disruption, reduce inflammation, increase keratinocyte proliferation, epithelialization, and collagen synthesis.	Indicated for acute and chronic wounds like ulcers and infected lesions.	[52]

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
