# Peer review of "An Overview of Wound Dressing Materials"

_pharmaceuticals, 2024, doi:10.3390/ph17091110_

Round 1

Reviewer 1 Report

Comments and Suggestions for Authors

The review paper is very interesting, the version is clear and comprise almost until the last advances in the area; very useful for the researchers and students working in the area. They have only   minor corrections to do.

Pag 6, line 278; What does it means Foci?  It would be necessary to explain it.

Pag 8, line 350, it would be also good to explain the word omentum flaps, in order to be understand for a wider audience.

Page 9, line 406, please correct cur cumin for curcumin.

Line 463, said gram positive and gram bacteria; is it gram-positive and gram-negative bacteria?

Page 467. It is missed one word in DINING ET AL [47] OLLOWED?

In section 3.4.5 Essential oils, line 479, it would be better to give some examples of them.

Line 518, the correct word is not popolis? It is propolis?

Author Response

Dear Reviewer 1

Thank you very much for taking the time to review this manuscript. Please find the detailed responses below and the corresponding revisions/corrections highlighted in the re-submitted files.

Comments 1: Pag 6, line 278; What does it means Foci?  It would be necessary to explain it.

Response 1: The sentence was rephased L309.

Comments 2: Pag 8, line 350, it would be also good to explain the word omentum flaps, in order to be understand for a wider audience.

Response 2: An explanation was added – L389-395

Comments 3: Page 9, line 406, please correct cur cumin for curcumin.

Response 3: Was corrected – L454

Comments 4: Line 463, said gram positive and gram bacteria; is it gram-positive and gram-negative bacteria?

Response 4: Yes. Was corrected – L514

Comments 5: Page 467. It is missed one word in DINING ET AL [47] OLLOWED?

Response 5: Followed. Was corrected – L517

Comments 6: In section 3.4.5 Essential oils, line 479, it would be better to give some examples of them.

Response 6: Examples were added L538-541

Comments 7: Line 518, the correct word is not popolis? It is propolis?

Response 7: Yes. Was corrected – L580

Reviewer 2 Report

Comments and Suggestions for Authors

With a giant article title, the manuscript conducted some popular science introductions with poor professional scientific meaning. Major revision is needed for being considered for publication in Pharmaceuticals.

1) At least 3 to 5 Figures and 1 Table please be added to make the contents more eye-drawing and more systematic.

2) Wound dressing has three key apparent issues, starting materials (https://doi.org/10.3390/biom14070789), creating techniques, and healing performances, the technique content should be added, such as popular electrospinning (10.13039/501100007129; https://doi.org/10.1039/D4TB00149D) and numerous nanotechnologies (https://doi.org/10.1002/wnan.1964;  https://doi.org/10.1002/adfm.202315020 ).

3) Perspective section can be added for pointing out the next step along the current state of this interdisciplinary region.

4) References formats are chaotic and lack some publication information, meanwhile, the most recent three years references can be more. To relate your work with the most recent developments can benefit a high impact of your article after publication.  

Author Response

Dear Reviewer 2

Thank you very much for taking the time to review this manuscript. Please find the detailed responses below and the corresponding revisions/corrections highlighted in the re-submitted files.

Comments 1: At least 3 to 5 Figures and 1 Table please be added to make the contents more eye-drawing and more systematic.

Response 1: A graphical abstract (L26-27), 3 Figures (L132-133, L176-177, L192-193) and a Table (L197-198) were added.

 Comments 2: Wound dressing has three key apparent issues, starting materials (https://doi.org/10.3390/biom14070789), creating techniques, and healing performances, the technique content should be added, such as popular electrospinning (10.13039/501100007129; https://doi.org/10.1039/D4TB00149D) and numerous nanotechnologies (https://doi.org/10.1002/wnan.1964;  https://doi.org/10.1002/adfm.202315020 ).

Response 2: We believe we respond to this comment by adding the text L528-534

The references were added. New references are highlighted.

Comments 3: Perspective section can be added for pointing out the next step along the current state of this interdisciplinary region.

Response 3: Section 4 has been changed (L609-630). We hope it meets the Reviewer expectations.

Comments 4: References’ formats are chaotic and lack some publication information, meanwhile, the most recent three years’ references can be more. To relate your work with the most recent developments can benefit a high impact of your article after publication. 

Response 4: References section was reviewed, and lacking information was added.

Most recent references were added, now 39 three years’ references are listed.

Reviewer 3 Report

Comments and Suggestions for Authors

The manuscript entitled "An Overview of Wound Dressing Materials” by Lagoa et al. is an interesting work and more valuable in medical sciences for various types of wounds. The author reviewed and addressed the wound dressing management for each kind of wound. The author discussed range of dressings materials from traditional, synthesis, and sponges, films, hydrogels, hydrocolloids, biological dressings such as collagen, gelatin, cellulose, bamboo, hyaluronic acid, sodium alginate, extracellular matrix bands, omentum flaps, autologous platelet-rich plasma, bioactive wound dressings such as curcumin, chitosan, xanthan gum, nanomaterials, essential oils, honey, and propolis. The author documents well with a high amount of literature support.

Hence, this manuscript is suitable for publication in pharmaceuticals journal. However, the author needs to address the following comments to improve the quality of manuscript further level

Comments to author

1.      The author needs to design an overview of the manuscript figure/scheme

2.      Suggested to include 4 more figures through design own or re-use from the published figure with reproduction permission

3.      The author should include future prospectives after the conclusion section based on experience in writing this manuscript and gaining knowledge from the literature

4.      A single table including all types of wound dressing materials related to the literature of good quality work needs to summarize

5.      The following literature is recommended to the author which may help make figures

6.       Selvaraj DhivyaViswanadha Vijaya Padma & Elango Santhini 

Wound dressing – an overview, BioMedicine, 2015, 5, 22

https://link.springer.com/article/10.7603/s40681-015-0022-9

7.       Luis J. BordaFlor E. Macquhae & Robert S. Kirsner

Wound dressings – a comprehensive review

Current Dermatology Reports 2016, 5, 287-297

https://link.springer.com/article/10.1007/s13671-016-0162-5

8.      Wound dressings: Current advances and future directions

Erfan Rezvani GhomiShahla KhaliliSaied Nouri KhorasaniRasoul Esmaeely NeisianySeeram Ramakrishna

Journal of Applied Polymer Science, 2019, 136 (27) 47738

https://onlinelibrary.wiley.com/doi/full/10.1002/app.47738

Author Response

Dear Reviewer 3

Thank you very much for taking the time to review this manuscript. Please find the detailed responses below and the corresponding revisions/corrections highlighted in the re-submitted files.

Comments 1: The author needs to design an overview of the manuscript figure/scheme

Response 1: A graphical abstract was added (L26-27).

Comments 2: Suggested to include 4 more figures through design own or re-use from the published figure with reproduction permission

Response 2: Three Figures were added, own design inspired by references (L132-133, L176-177, L192-193)

Comments 3: The author should include future prospectives after the conclusion section based on experience in writing this manuscript and gaining knowledge from the literature

Response 3: Section 4 has been changed (L609-630). We hope it meets the Reviewer expectations.

Comments 4: A single table including all types of wound dressing materials related to the literature of good quality work needs to summarize

Response 4: A Table was added (L197-198).

Comments 5: The following literature is recommended to the author which may help make figures

Selvaraj Dhivya, Viswanadha Vijaya Padma & Elango Santhini

Wound dressing – an overview, BioMedicine, 2015, 5, 22

https://link.springer.com/article/10.7603/s40681-015-0022-9

Luis J. Borda,  Flor E. Macquhae & Robert S. Kirsner

Wound dressings – a comprehensive review

Current Dermatology Reports 2016, 5, 287-297

https://link.springer.com/article/10.1007/s13671-016-0162-5

Wound dressings: Current advances and future directions

Erfan Rezvani Ghomi, Shahla Khalili, Saied Nouri Khorasani, Rasoul Esmaeely Neisiany, Seeram Ramakrishna

Journal of Applied Polymer Science, 2019, 136 (27) 47738

https://onlinelibrary.wiley.com/doi/full/10.1002/app.47738

Response 5: Thank you for the advice. Three Figures were added, own design inspired by references (L132-133, L176-177, L192-193).

The references were added. New references are highlighted.

Round 2

Reviewer 2 Report

Comments and Suggestions for Authors

Reviewer thanks to all authors for their efforts to answer all questions and making necessary revisions. The manuscript’s quality has been substantially improved and the responses are clear and convinced. I recommend its acceptance for publication in its present form.

Author Response

Dear Reviewer 2

The authors wish to sincerely thank Reviewer 2 for his valuable contributions by revising the paper and for the constructive suggestions. We agree that the paper has been substantially improved.

Best regards

Cristina Queiroga